# Mechanism of lateral cell-wall expansion at a constant diameter in *Bacillus subtilis*

Yucheng Liang [1], Laure Bellard [2], Yung-Sing Wong [3], Cécile Morlot [2], Jean-Emmanuel Hugonnet [1] ✉, Filippo Rusconi [1,4,5] ✉ & Michel Arthur [1,5] ✉

In *Escherichia coli*, lateral cell-wall expansion during growth occurs by cross-linking of new glycan strands to the existing peptidoglycan network. However, it is unclear whether the same mechanism applies to other rod-shaped bacteria. Here, we use cell imaging and mass spectrometry analysis of isotopically labeled peptidoglycan to study this process in the Gram-positive bacterium *Bacillus subtilis*. We show that new glycan strands are cross-linked exclusively to other newly synthesized glycan chains, and not to the existing peptidoglycan network. We propose that new peptidoglycan meshes, assembled at the membrane surface, impose a shift on older meshes toward the bacterial surface, where they sustain the cytoplasmic turgor pressure, before their eventual degradation. This outward movement would result in preferential lateral expansion due to the large difference in the strain tensors of the peptidic and glycosidic peptidoglycan components.

The cytoplasmic membrane of bacterial cells is surrounded by a giant peptidoglycan (PG) macromolecule ($10^{10}$ to $10^{11}$ Da) that mechanically counteracts the turgor pressure of the cytoplasm during the entire cell cycle and determines bacterial cell shape[1]. PG is a heteropolymer consisting of glycan strands linked to each other by short peptides. The glycan moiety is made of alternating β−1,4-linked *N*-acetylglucosamine (GlcNAc) and *N*-acetylmuramic acid (MurNAc) residues. The peptide consists of L-Ala[1]-γ-D-Glu[2]-DAP[3]-D-Ala[4]-D-Ala[5] in which DAP is diaminopimelic acid or its amidated form in *Escherichia coli* and *Bacillus subtilis*, respectively[1]. The stem peptides from adjacent glycan strands are linked to each other in the periplasm by the formation of 4 → 3 cross-links connecting D-Ala[4] to DAP[3]. Formation of these cross-links is catalyzed by D,D-transpeptidases, also referred to as penicillin-binding proteins (PBPs), which are the essential targets of β-lactam antibiotics. In *E. coli*, about half of the PG is turned over and recycled at each generation (Supplementary Fig. 1A). The specific PG moieties that are recycled in the PG assembly pathway are the tripeptide L-Ala[1]-γ-D-Glu[2]-DAP[3] and the glucosamine moiety of GlcNAc and MurNAc. In *B. subtilis*, the disaccharide-peptide subunit undergoes a series of maturation reactions leading to the trimming of either or both D-Ala residues, the

amidation of DAP, and the *N*-deacetylation of GlcNAc residues[2] (Supplementary Fig. 1B).

In the past two decades, a wealth of PG labeling techniques and imaging methods has been developed to enable the description of the sites and timing of PG synthesis in various bacterial cells, thus revealing different modes of PG expansion (dispersed or zonal insertion, at lateral, midcell, or polar sites), including different modes for bacteria of similar shape[3–6]. At the cellular scale, imaging based on incorporation of fluorescent D-Ala or clickable D-Ala-D-Ala dipeptides has revealed similar modes of PG expansion in rod-shape *E. coli* and *B. subtilis* cells, with dispersed insertion of newly synthesized PG in the lateral wall. In contrast, the side wall of *Mycobacterium tuberculosis* and *Agrobacterium tumefaciens* expands from an annular zone located at the proximity of the new cell's pole[3]. Zonal expansion of the side wall is also observed in ovococci, such as *Streptococcus pneumoniae*, in which PG is synthesized within an annular region at midcell. Using direct stochastic optical reconstruction microscopy (dSTORM), PG expansion in *S. pneumoniae* was further shown to proceed through insertion of new material at septal cleavage sites located at the periphery of the septum[5,7].

[1]INSERM ERL 1336, UMR 8228, Sorbonne Université-ENS-PSL-CNRS, Paris, France. [2]Univ. Grenoble Alpes, CNRS, CEA, IBS, Grenoble, France. [3]Univ. Grenoble Alpes, CNRS, DPM, Grenoble, France. [4]GQE-Le Moulon/PAPPSO, IDEEV, Université Paris-Saclay, INRAE, CNRS, AgroParisTech, Gif-sur-Yvette, France. [5]These authors contributed equally: Filippo Rusconi, Michel Arthur. ✉e-mail: jean-emmanuel.hugonnet@crc.jussieu.fr; filippo.rusconi@universite-paris-saclay.fr; michel.arthur@crc.jussieu.fr

Since bacterial cell imaging does not provide access to the mode of PG synthesis at the submolecular level, much is still unknown about how PG chemical bonds are formed and cleaved in the processes leading to side wall expansion. Recently, we have shown in *E. coli* that high-resolution mass spectrometry analyses of PG labeled with the [$^{13}$C] and [$^{15}$N] heavy isotopes of carbon and nitrogen can be used to determine the mode of insertion of newly synthesized glycan strands into the existing PG at the atomic level[8]. This approach showed that the glycan strands are incorporated one-at-a-time between two existing glycan strands by the concerted action of transpeptidases, which cross-link newly synthesized strands to existing PG, and of endopeptidases, which act as space-making enzymes by cleaving existing cross-links (Supplementary Fig. 2).

In this work, we used a combination of imaging and mass spectrometry approaches to compare the modes of PG expansion in *E. coli* and *B. subtilis*. We show that the disperse mode of insertion of newly synthesized PG into the lateral wall, as detected by cell imaging for both bacterial species, involves radically different modes of PG synthesis at the molecular level.

## Results

### The PG is extensively turned over in *Bacillus subtilis*

The PG of *B. subtilis* strain 168 was fully labeled with stable isotopes of carbon and nitrogen upon growth for at least 10 generations. The growth medium contained a bacterial extract prepared by sonicating *E. coli* cells grown in a labeled minimal medium, followed by enzymatic digestion of the resulting lysate with proteases, DNAses, and RNAses to generate a nutrient-rich homogenate (see online methods). At an optical density at 600 nm of *ca.* 0.4, bacteria were further grown in unlabeled medium, and samples were collected 5, 10, 15, and 20 min after this medium switch. PG sacculi were prepared from each sample, digested with muramidases, and the resulting disaccharide-peptide fragments were purified by *rp*HPLC and characterized by high-resolution mass spectrometry (MS). Analysis of the main monomer, the GlcNAc-MurNAc-tetrapeptide, revealed effective (*ca.* 98%) labeling with both the [$^{13}$C] and [$^{15}$N] nuclei prior to the medium switch (time = 0 min) (Fig. 1a). After the medium switch, the heavy isotopologues were gradually replaced by the unlabeled ([$^{12}$C] and [$^{14}$N]) isotopologues.

The heavy and light isotopologues of the GlcNAc-MurNAc-tetrapeptide muropeptide were present in the same amount 13.4 ± 3.2 min after the medium switch, as shown by the time at which the red (% of labeled) and purple (% of unlabeled) curves intersect (Fig. 2a for representative kinetics, Table 1, and Supplementary Table 1 for the complete set of data). The time at which such curves intersect will hereinafter be referred to as the $t_{50\%}$ value, which will be consistently used throughout this report as a quantitative parameter to compare the kinetics of replacement of existing material by newly synthesized material in the PG. In the absence of any turnover of the PG, equal amounts of labeled and unlabeled isotopologues are expected to be present after one generation time (expected $t_{50\%}$ = one generation time i.e., 36.1 ± 4.6 min under the experimental growth conditions). Indeed, the amount of existing (labeled) PG is expected to be stable over time in the absence of any turnover, whereas the total amount of PG doubles in one generation, leading to the formation of an equivalent amount of unlabeled (light) PG. Thus, the kinetics of isotopologue replacement reveal that the PG of *B. subtilis* exhibits turnover because the observed $t_{50\%}$ value is shorter than the generation time (13.4 ± 3.2 min *versus* 36.1 ± 4.6 min, respectively). In this study, PG turnover is defined as the degradation of the polymer by the combined action of lytic enzymes that cleave glycosidic and amide bonds, referred to hereinafter as autolysins[9]. This degradation is compensated by neo-synthesis in excess of that required for the net increase in PG in the growing culture. A numeric modeling of these concepts is presented in Supplementary Data 1, indicating that an

exponential decay constant of 0.0266 min$^{-1}$ (half-life of 26.06 min) can be fitted to the observed $t_{50\%}$ of 13.4 min. Accordingly, it can be estimated that *ca.* 69% of the PG is degraded in one generation. This turnover is more extensive than that reported for *E. coli* (40–50%)[10–13].

### PG turnover is not associated with recycling

In *E. coli*, half of the disaccharide-peptide units is released from the PG at each generation by autolysins[11,12]. About 95% of this material is recycled both as glucosamine-6P (a common precursor for GlcNAc and MurNAc synthesis) and as the L-Ala-γ-D-Glu-DAP tripeptide (directly added to UDP-MurNAc)[11,13,14]. These recycling pathways (Supplementary Fig. 1A) result in hybrid isotopologues containing heavy glucosamine and tripeptide moieties[8] (isotopologues h1 and h2 in the mass spectrum shown in Fig. 1b). In *B. subtilis*, no isotopologue generated by the recycling of glucosamine or L-Ala-γ-D-Glu-DAP$_{NH2}$ was detected in the mass spectra (Fig. 1a). Thus, no recycling pathway analogous to that existing in *E. coli* was detected in *B. subtilis*.

### Heavy D-Glu in PG originates from existing metabolite pools

Mass spectra obtained for *B. subtilis* revealed hybrid isotopologues that were not observed for *E. coli*. In comparison to the unlabeled isotopologues, these hybrid isotopologues had mass increments corresponding to the presence of 5 or 6 heavy nuclei (Fig. 1a). Isotopologues with these mass increments were detected for all muropeptides (Supplementary Data 2). The localization of the heavy nuclei accounting for the +5 and +6 mass increments was determined by tandem mass spectrometry (MS/MS) and labeling of PG with [$^{15}$N] only (Supplementary Fig. 3, 4). This analysis revealed that the +5 increment is due to the presence of D-Glu residues containing [$^{13}$C] isotopes at all of its five carbon positions, whereas the +6 mass increments correspond to D-Glu residues additionally containing a [$^{15}$N] nitrogen nucleus. The latter isotopologue might result from the presence of an abundant pool of existing (heavy) L-Glu that would persist after the medium switch and be converted to D-Glu by the RacE and YrpC glutamate racemases[15,16]. The isotopologue with the +5 mass increment might result from the presence of an abundant 2-oxoglutarate pool that would be converted to D-Glu by D-alanine-2-oxoglutarate aminotransferase using D-Ala as the source of unlabeled NH$_3$[17]. Thus, the isotopologues containing labeled D-Glu may originate from existing L-Glu and 2-oxoglutarate pools rather than from the selective recycling of D-Glu from the existing PG. The fast replacement of heavy by light isotopes observed for the +5 and +6 mass increment isotopologues are in agreement with this hypothesis (Supplementary Fig. 5).

### PG subunit polymerization and maturation are concomitant

The PG precursor subunit of *B. subtilis*, as assembled in the cytoplasm, consists of a pentapeptide (Penta) stem (L-Ala-γ-D-Glu-DAP$_{NH2}$-D-Ala-D-Ala) attached to the MurNAc residue of the GlcNAc-MurNAc disaccharide. Structural diversity in the disaccharide-peptide subunits is introduced in the periplasm by the activity of (i) D,D-carboxypeptidases and L,D-carboxypeptidases that cleave the D-Ala-D-Ala and DAP$_{NH2}$-D-Ala bonds, thus generating tetrapeptide (Tetra) and tripeptide (Tri) stems, respectively, (ii) deaminases that convert DAP$_{NH2}$ into DAP, and (iii) deacetylases that remove the acetyl group of GlcNAc residues (Supplementary Fig. 1B). Kinetic analysis of the relative abundance of isotopologues, as described above for the disaccharide-tetrapeptide, showed that the $t_{50\%}$, i.e. the time at which the labeled and unlabeled isotopologues are present in equal amounts, was similar for all monomers (Fig. 2a for representative kinetics, Table 1, and Supplementary Table 1 for the complete set of data). Thus, the maturation reactions occur concomitantly with PG polymerization. The opposite behavior prevails in *E. coli* as the curves for the tetrapeptide and tripeptide intersect at different times ($t_{50\%}$ = 33.8 min *versus* 48.6 min, respectively; Fig. 2b). Thus, the production of

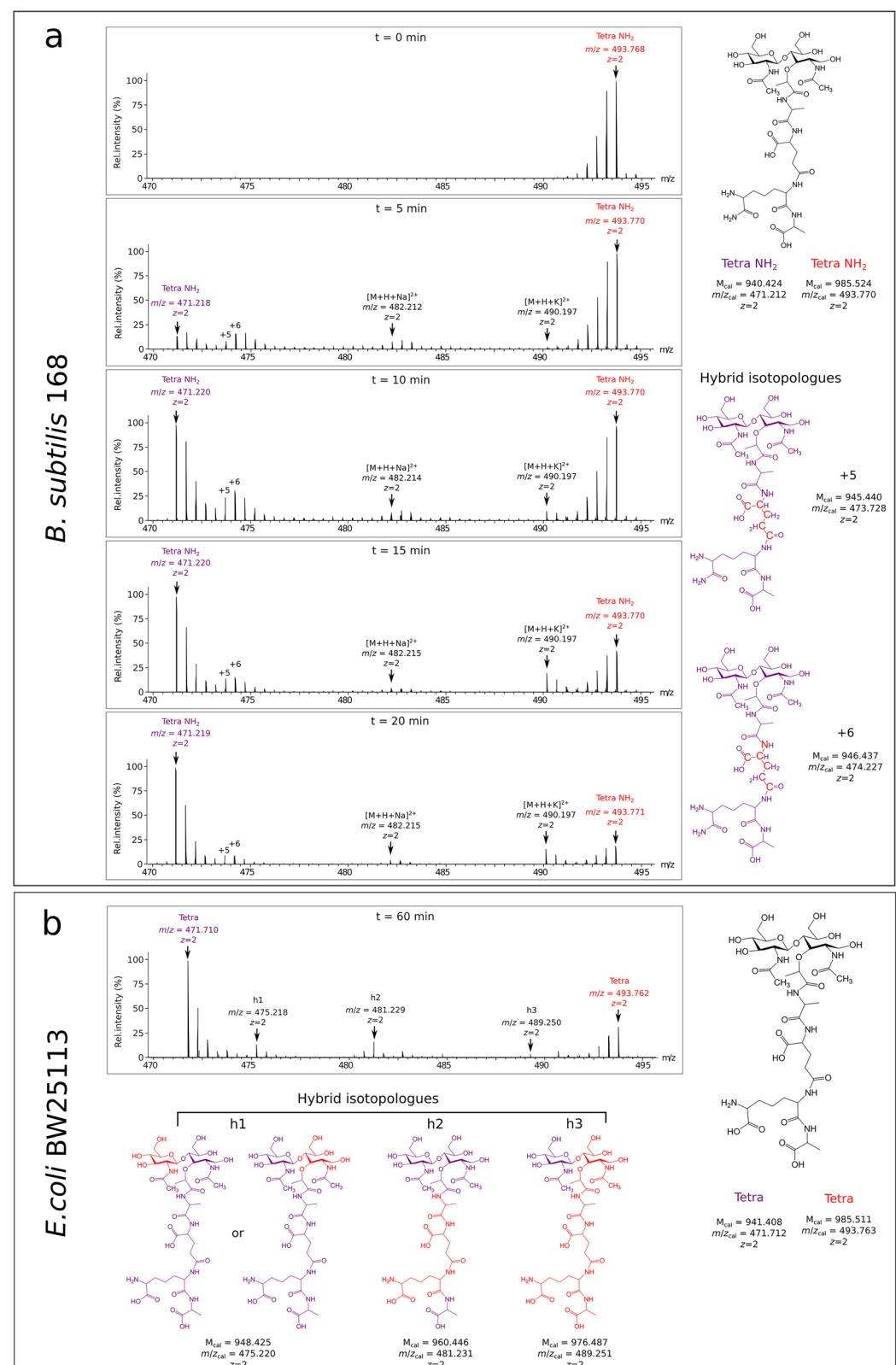

**Fig. 1 | Mass spectra showing the labeled, unlabeled, and hybrid isotopologues.**
**a** Isotopologues of the reduced GlcNAc-MurNAc-L-Ala-γ-D-Glu-DAP_NH2-D-Ala (Tetra NH₂) muropeptide of *B. subtilis* strain 168. Spectra were obtained for the Tetra NH₂ muropeptide purified from PG extracted immediately before the medium switch (t = 0 min) or 5, 10, 15, and 20 min after the medium switch. The structure of the +5 and +6 hybrid isotopologues was determined by tandem mass spectrometry (see below and Supplementary Fig. 3 and 4). The three types of isotopologues, labeled, unlabeled, and hybrid correspond to molecules exclusively containing [¹³C] and [¹⁵N] atoms, exclusively containing [¹²C] and [¹⁴N] atoms, or containing combinations of heavy and light isotopes, respectively. **b** Isotopologues of the reduced GlcNAc-MurNAc-L-Ala-γ-D-Glu-DAP-D-Ala tetrapeptide (Tetra) from *E. coli* strain BW25113, which differs from that of *B. subtilis* by the absence of amidation of the side-chain carboxyl group of DAP. Color code: red, labeling with heavy [¹³C] and [¹⁵N] isotopes; purple, no labeling ([¹²C] and [¹⁴N] isotopes). Spectra are representative of one out of four biological repeats. Data for *E. coli* are from ref. 8.

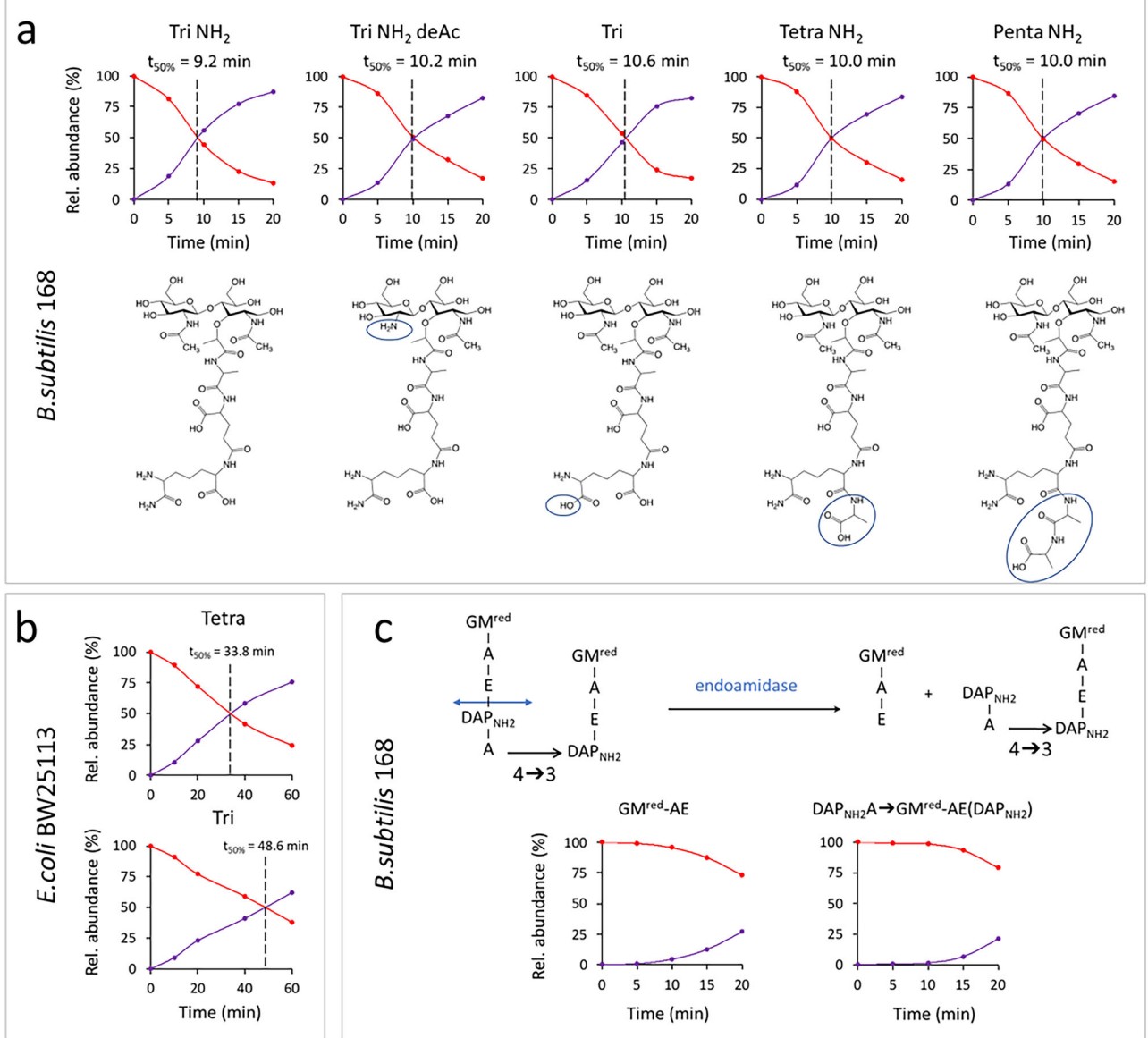

**Fig. 2 | Kinetics of the replacement of heavy by light isotopologues in muropeptides originating from PG maturation. a** Monomers from *B. subtilis* strain 168. Differences between Tri NH$_2$ and other muropeptides are circled in blue. Dashed lines point to the time at which the relative amount of existing (red) and newly synthesized (purple) muropeptide isotopologues are equal ($t_{50\%}$). **b** Tetrapeptide and tripeptide monomers from *E. coli* strain BW25113. **c** Products of cleavage of the Tetra→Tri dimer following cleavage of the D-Glu-DAP bond by an endoamidase (double headed blue arrow). Color code: red, heavy isotopes; purple,

light isotopes. Kinetics data are from one representative experiment out of three biological repeats ($n = 3$). The full data set, including the means and standard deviations for the three repeats, appears in Table S1. tri tripeptide, tetra tetrapeptide, penta pentapeptide, NH$_2$ amidation of DAP, deAc deacetylation of GlcNAc, GMred reduced GlcNAc-MurNAc, A L-Ala or D-Ala, E D-Glu, DAP$_{NH2}$, amidated diaminopimelic acid. The generation times were $36 \pm 5$ and $67 \pm 1$ for *B. subtilis* and *E. coli*, respectively.

tripeptide starting from tetrapeptide is coupled to PG polymerization in *B. subtilis* but not in *E. coli*.

## Cleavage of dimers is not coupled to PG polymerization

The products of the cleavage of the amide bond connecting D-Glu to DAP$_{NH2}$ of the cross-linked Tetra→Tri dimer are shown in Fig. 2c. Formation of these products displayed similar kinetics in agreement with their common origin. For both products, the replacement of the labeled by the unlabeled isotopologue was very slow, with $t_{50\%}$ values much greater than 20 min. These results indicate that the endolytic cleavage is not coupled to the formation of the Tetra→Tri dimer ($t_{50\%}$ of $16.4 \pm 4.0$ min; Table 1) but corresponds to a PG maturation that is delayed with respect to PG cross-linking. The γ-D-Glu-DAP$_{NH2}$ endolytic cleavage might therefore potentially contribute to the

separation of the daughter cells after the completion of septum synthesis. Accordingly, separation of *B. subtilis* daughter cells was reported to predominantly rely on the hydrolysis of the γ-D-Glu-DAP$_{NH2}$ amide bond, whereas hydrolysis of the MurNAc-L-Ala bond had a relatively minor role in this process[18].

## PG cross-linking and glycan chain assembly are decoupled

In *E. coli*, PG polymerization is performed by two specialized multi-enzyme complexes, the elongasome and the divisome, for lateral expansion of the wall and synthesis of the septum, respectively[19,20]. These complexes contain both glycosyltransferases and monofunctional transpeptidases, namely RodA and PBP2 for the elongasome, and FtsW and PBP3 for the divisome[16,20]. PG polymerization is also performed by bifunctional PBP1a and PBP1b that combine both

**Table 1 | Values of $t_{50\%}$ for representative monomers ($n = 5$) and dimers ($n = 5$)**

| Monomers[a] | Average[c] | SD[c] |
|---|---|---|
| Tri NH$_2$ | 11.9 | 2.6 |
| Tetra NH$_2$ | 13.4 | 3.2 |
| Penta NH$_2$ | 12.6 | 2.8 |
| Tri NH$_2$ deAc | 13.7 | 2.8 |
| Tri | 12.4 | 2.2 |
| All monomers | 12.8 | 0.7 |
| **Dimers[b]** | | |
| Tetra→Tri (NH$_2$)$_2$ | 16.4 | 4.0 |
| Tetra→Tetra (NH$_2$)$_2$ | 19.3 | 5.2 |
| Tetra→Penta (NH$_2$)$_2$ | 14.7 | 4.3 |
| Tetra→Tri (NH$_2$)$_1$ | 14.0 | 2.9 |
| Tetra→Tri deAc (NH$_2$)$_2$ | 17.4 | 3.9 |
| All dimers | 16.4 | 2.1 |
| **Anhydro muropeptides** | | |
| Tri NH$_2$ | 16.1 | 3.2 |
| Tetra→Tri (NH$_2$)$_2$ | 18.1 | 4.7 |

[a]The monomers contained amidated DAP (NH$_2$) and tripeptide, tetrapeptide, or pentapeptide stems. The remaining dimer contained a tripeptide stem and glucosamine instead of *N*-acetylglucosamine (deAc) or DAP instead of amidated DAP.

[b]The dimers contained a tetrapeptide stem in the donor position and a tripeptide, tetrapeptide, or pentapeptide stem at the acceptor position. The amidation status of DAP residues is specified with NH$_2$. One dimer contained glucosamine instead of *N*-acetylglucosamine (deAc).

[c]Average $t_{50\%}$ values were calculated from four independent experiments ($n = 4$; SD, standard deviation). The $t_{50\%}$ values were defined as the time at which the relative amounts of labeled and unlabeled muropeptide isotopologues are equal (see Fig. 2).

glycosyltransferase and transpeptidase catalytic domains within the same polypeptide chain. In vitro polymerization assays using purified enzymes and PG precursors have shown that the colocalization of glycosyltransferase and transpeptidase catalytic domains, within either the same protein complex or the same polypeptide chain, is associated with the obligate coupling of the corresponding reactions[21–24]. The existence of a similar coupling in *B. subtilis* would imply that monomers and cross-linked dimers should be concomitantly synthesized, and hence similar $t_{50\%}$ values should be observed for the two types of muropeptides. In contrast, if glycan chains were first polymerized and secondarily cross-linked, the $t_{50\%}$ values should be higher for dimers than for monomers. Kinetics analyses revealed homogeneous $t_{50\%}$ values for the dimer variants generated by the maturation reactions discussed above for the monomers (Fig. 3a for representative kinetics, Table 1, and Supplementary Table 2 for the complete set of data). Figure 3b shows that there is a statistically significant difference between the $t_{50\%}$ of dimers and monomers ($p = 0.0001$; one sample Wilcoxon non-parametric test) (Fig. 3b and Supplementary Table 3). These results indicate that the polymerization of glycan chains and the formation of the cross-links are not concomitant in *B. subtilis*, and thus, that transglycosylation and transpeptidation reactions are, at least in part, not coupled in this bacterium.

### Glycan chain synthesis is completed prior to cross-linking

In *E. coli*, an anhydro-MurNAc residue is present at the reducing end of glycan chains of mature PG, resulting from the formation of an internal 1,6 glycosidic bond by lytic transglycosylases[25,26]. The anhydro-MurNAc residue can be formed in a reaction that releases newly synthesized glycan chains from the lipid carrier undecaprenyl-phosphate[27]. Anhydro-MurNAc residues can also result from the cleavage of glycosidic bonds in mature PG. Kinetic analyses performed in *B. subtilis* revealed that the $t_{50\%}$ value is greater for the anhydro than for the reduced forms of the tripeptide monomer (16.1 ± 3.2 *versus*

11.9 ± 2.6) (Fig. 3c for representative kinetics, Table 1, and Supplementary Table 4 for the complete set of data). In contrast, similar values were observed for the anhydro muropeptide monomer and all reduced dimers (16.1 ± 3.2 *versus* 16.4 ± 2.1). Thus, the release of the glycan strands from the lipid carrier is concomitant with the cross-linking reaction.

### Cross-links only involve newly synthesized stems

In *E. coli*, the expansion of the lateral wall proceeds by the insertion of a single newly synthesized glycan chain between two existing glycan chains (Supplementary Fig. 2). This one-at-a-time mode of PG expansion implies that newly synthesized glycan strands are exclusively cross-linked to existing stems, leading to the formation of hybrid isotopologues of both the Tetra→Tetra and Tetra→Tri dimers. Since pentapeptide stems that are not rapidly used as acyl donors by D,D-transpeptidases are eliminated from the existing PG by D,D-carboxypeptidases, the hybrid isotopologues exclusively contain a newly synthesized acyl donor and an existing acceptor (isotopologues of the new→old type)[8,28]. In contrast, newly synthesized glycan chains are cross-linked to each other in the septum, thus generating new→new isotopologues. In this study, we show that the new→old isotopologues, which are preponderant in *E. coli*, are not present in the PG of *B. subtilis* (Supplementary Data 2). These results indicate that newly synthesized cross-linked dimers are assembled in the periplasm of *B. subtilis* in the absence of formation of any cross-link between newly synthesized and existing PG subunits.

### Disperse incorporation of newly synthesized glycan strands

To locate the sites of PG expansion by cell imaging, we synthesized an analog of the D-Ala-D-Ala PG precursor containing an ethinyl group at the N-terminal position. The ethinyl-D-Ala-D-Ala molecule was incorporated into cytoplasmic PG precursors in lieu of a small fraction of the endogenous D-Ala-D-Ala, resulting in the presence of an ethinyl group at the 4$^{th}$ position of stem peptides in the newly synthesized PG. Regions of active cell wall synthesis were identified by fluorescence microscopy following conjugation of azido-Alexa Fluor 532 to the ethinyl-D-Ala residue by a copper-catalyzed click chemistry reaction. In agreement with previous analyses[1,3,4], the observation of fluorescent signal along the cylindrical part of the cell indicates that lateral cell wall expansion involves disperse PG synthesis in *B. subtilis* (Fig. 4).

## Discussion

In this study, mass spectrometry analysis of isotopically labeled PG reveals drastically distinct modes of PG assembly in *E. coli* and *B. subtilis*. The products of PG turnover are recycled into the PG synthesis pathway in *E. coli* but not in *B. subtilis* (Fig. 1). Maturation of the PG subunit is concomitant with glycan chain polymerization in *B. subtilis*, whereas maturation reactions occur later in *E. coli* (Fig. 2). Glycan chain polymerization and PG cross-linking are thought to be coordinated in *E. coli*, whereas in *B. subtilis* glycan strands appear to be fully polymerized prior to their release from the lipid carrier and cross-linking. This conclusion is supported by comparing the delays in the replacement of existing by newly synthesized PG material. Indeed, these delays are greater for dimers than for monomers, but similar for dimers and anhydro muropeptides (Table 1 and Fig. 3). In addition, the mode of insertion of newly synthesized PG into the expanding side wall of *B. subtilis* drastically differs from that previously reported for *E. coli*[8,28]. Indeed, the isotopic composition of dimers indicates that newly synthesized glycan chains are exclusively cross-linked to newly synthesized glycan chains in *B. subtilis* (new→new isotopologues), but predominantly to existing PG in *E. coli* (new→old isotopologues). Thus, the one-at-a-time mode of insertion of newly synthesized glycan strands into the side wall PG that prevails in *E. coli* is not operating in *B. subtilis*. Zonal expansion of the PG in the peripheral area of the septum, as described in *S. pneumoniae*, might account for the cross-linking of

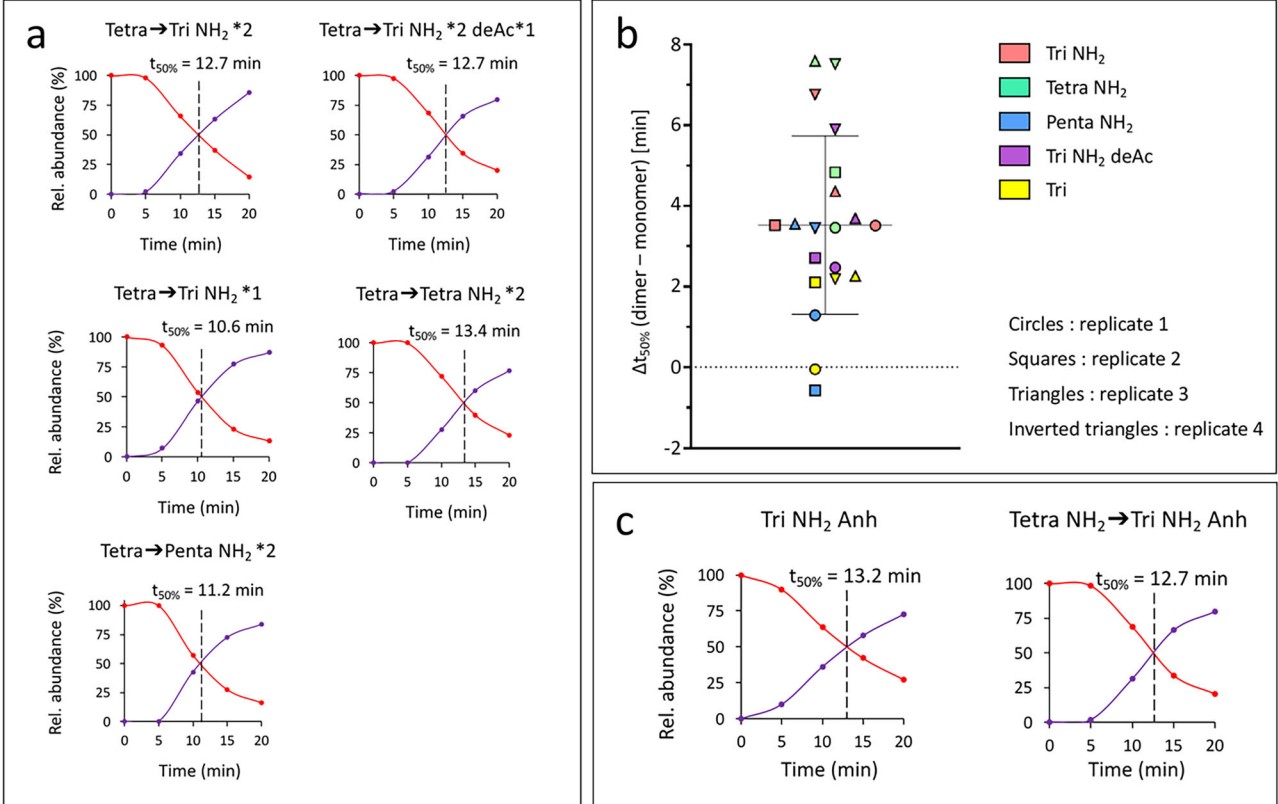

**Fig. 3 | Delay between the synthesis of monomers and that of dimers and anhydro-muropeptides in *B. subtilis* strain 168. a** Determination of $t_{50\%}$ values for five muropeptide dimers. The arrows represent 4 → 3 cross-links formed by the D,D-transpeptidase activity of the PBPs. Polymorphism in muropeptides included the presence or absence of amidation ($NH_2$) and the deacetylation of GlcNAc (deAc). Tri, tripeptide; Tetra, tetrapeptide; Penta, pentapeptide. **b** Distribution of the delays for 5 matched monomer-dimer pairs, that were matched according to both the nature of the free stem (tripeptide, tetrapeptide, or pentapeptide) and the acetylation or amination status of free tripeptide stems (see colored symbols and corresponding legend). As an example, the orange symbol corresponds to the difference between the $t_{50\%}$ observed for the Tri $NH_2$ monomer and the Tetra $NH_2$ → Tri $NH_2$ dimer. Data are the means of 4 biological replicates ($n = 4$). Horizontal lines represent the mean. The vertical whisker represents the standard deviation. The full set of data appears in Supplementary Table 3. **c** Kinetic analysis of the relative abundance of anhydro-muropeptides from *B. subtilis* strain 168. Color code: red, heavy isotopes; purple, light isotopes. Kinetics data are from one representative experiment out of three biological repeats. The full data set, including the means and standard deviations for the three repeats ($n = 3$), appears in Supplementary Table 4.

newly synthesized glycan chains between each other. However, imaging of the sites of PG synthesis based on metabolic labeling excludes this possibility because this approach shows a disperse incorporation of new material into the side wall (Fig. 4 and refs. 1,3,4).

Because neither the one-at-a-time nor the zonal mode of PG expansion is compatible with the results obtained in this study, we envision the possibility that the side wall be made of concentric rod-shape meshes of PG that are not linked to each other (Fig. 5a, b). According to this model, newly synthesized meshes are constantly assembled at the outer face of the cytoplasmic membrane, forcing older meshes to shift toward the bacterial surface, where they are eventually degraded. The whole process is analogous to desquamation. In agreement with MS analyses, which revealed the exclusive synthesis of isotopologues of the new→new type, the desquamation model implies that cross-link formation is restricted to peptide moieties carried by newly synthesized glycan strands in the innermost PG layer undergoing synthesis at the surface of the membrane (purple layer in the right panel of Fig. 5b). This mode of PG synthesis is also consistent with the fluorescence microscopy data because it implies that the sites of PG synthesis are evenly distributed over the whole surface of the bacterium.

In *E. coli*, the one-at-a-time mode of insertion of newly synthesized glycan strands, as well as concomitant glycan chain polymerization and peptide stem cross-linking reactions, can account for the expansion of the lateral wall at a constant diameter[28]. Indeed, cross-linking of

a newly synthesized glycan strand to two existing glycan strands followed by the cleavage of the cross-links connecting the two existing strands is expected to result in the lateral expansion of the cell wall (Supplementary Fig. 2). In this process, the existing PG may act as a template for the assembly of new material, enabling to maintain a constant diameter[29,30]. In *B. subtilis*, the desquamation model, *i.e.* the continuous synthesis of PG layers at the surface of the cytoplasmic membrane, coupled to the degradation of the outermost PG layers, might also account for the expansion of the side wall at a constant diameter, but by an entirely different mechanism that is described in the following two paragraphs.

The desquamation model proposes that the turgor pressure of the cytoplasm is sustained by the outermost PG layers, such that the glycan and interpeptide bridge components of those layers are stretched and adopt their most extended conformation (Fig. 5). In contrast, the glycan and peptide components of the newly synthesized layers at the surface of the membrane are not expected to be constrained and would therefore adopt less extended conformations. Thus, the shift of the PG layers toward the cell periphery would result in an expansion of the cell surface.

The desquamation model also accounts for the expansion of the side wall at a constant diameter. Although the turgor pressure of the cytoplasm applies isotropically onto the PG bonds, the outward shift of the PG layers promotes an asymmetric expansion of the lateral cell wall, as detailed below. Because the turgor pressure is isotropic, it exerts an

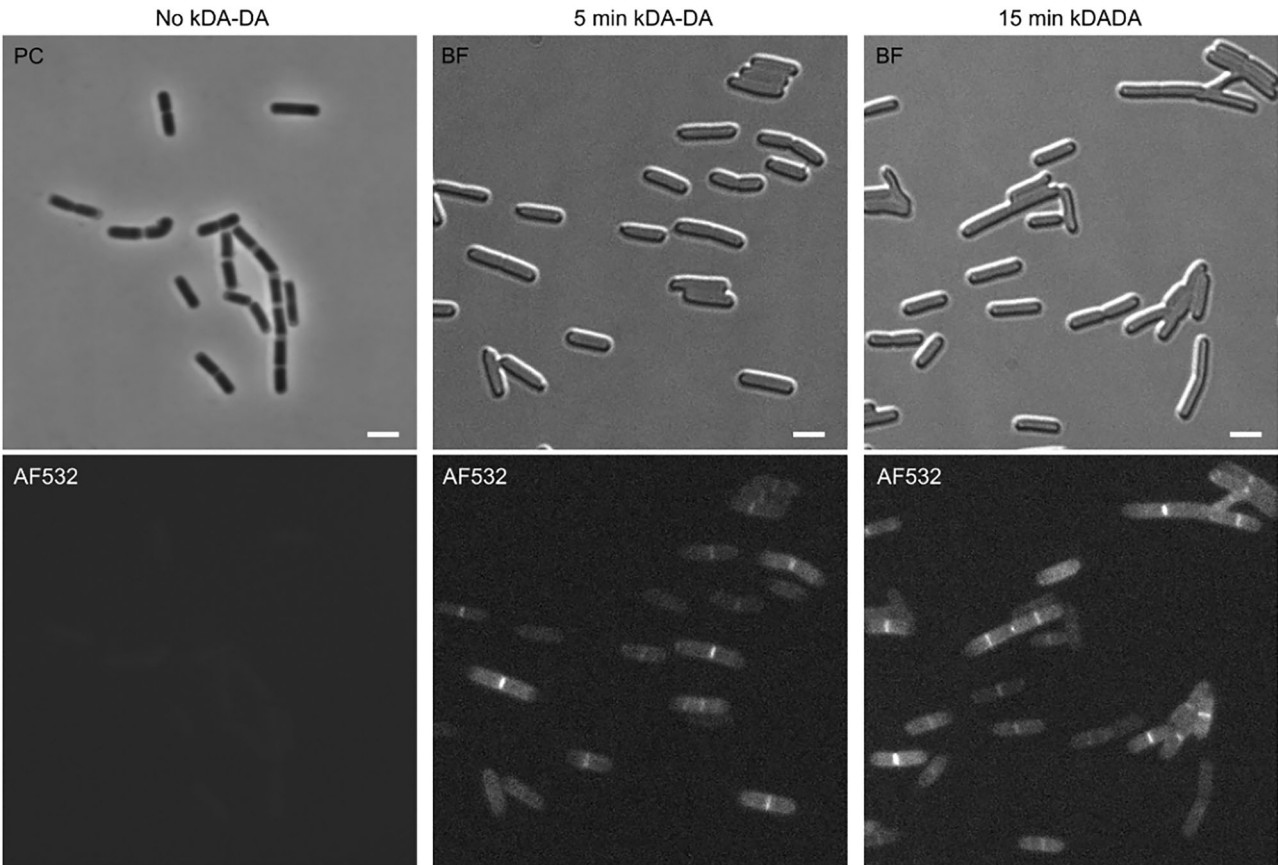

**Fig. 4 | Localization of newly synthesized peptidoglycan in *B. subtilis* cells.** Fields of *B. subtilis* strain 168 observed by fluorescence (lower panels), phase contrast (PC) or bright field (BF) microscopy. Cells in exponential growth phase were grown for 15 min in the presence of ethinyl-D-Ala-D-Ala or in the absence of the probe, fixed, and incubated with azido-AF532 for fluorescent labeling by copper-catalyzed click chemistry. Scale bars, 2 μm. The images are representative of 3 independent experiments.

equivalent force onto both the glycosidic bonds and the interpeptide bridges, which are oriented perpendicularly to and along the long axis of the cell, respectively (Fig. 5a). This should result in a deformation of the PG mesh in both directions, the amplitude of which is described as the strain tensor, a physical dimension measuring the deformation per unit length at each point in a body subjected to a force. The lateral extension of the side wall could be accounted for by the large strain tensor of the peptide moiety, resulting from the presence in each interpeptide bridge of a total of eight amide bonds and eight methylene groups, as supported by analyses of the physical properties of the PG and by the determination of the structure of the PG fragments resolved in solution by NMR[31–33]. The strain tensor of the glycosidic moiety, being much lesser, accounts at most for a marginal increase in the diameter of the expanding side wall, or, potentially, for no increase at all. Indeed, the strain tensor of the glycosidic moiety might not integrally translate into a net increase in the diameter of the bacterial cell because of the very small diameter difference of the inner and outer PG layers (in the order of 40 nm for a cell diameter of 800 nm, which implies a difference in the PG layer circumference of 5%[1]).

As described in Fig. 5b, the desquamation model implies that the number of glycan strands incorporated in a newly synthesized PG layer at the surface of the cytoplasmic membrane be greater than that of the strands present in older layers (compare layers V and VI in the left and right panels of Fig. 5b). The combined effect of this greater number of glycan strands and of the stretching of the interpeptide bridges, upon the outward shift of the PG layers, accounts for the lateral cell wall expansion. This outward shift is only possible if the outermost layer is degraded by autolysins. In agreement with this requirement, our experimental data show that the amount of PG degraded in one generation equals 69% of the total amount of PG present at the end of that generation. Much lower turnover values (8–10%) have been reported in pulse chase experiments involving the determination of radioactive PG material released in the external medium in the first minutes of the chase. However, monitoring the release of PG material on a longer time scale revealed that turnover increases sixfold to reach 50–60% after 1–2 generations[34]. This result, along with kinetics obtained with externally added autolysins, led to the conclusion that PG turnover is delayed because labeled PG is not accessible to autolysins at the beginning of the chase period[34]. The observation of this delay is in full agreement with our desquamation model since radioactive material assembled at the surface of the cytoplasmic membrane during the pulse needs to migrate to the outermost layer so as to become accessible to externally added autolysins. Of note, the PG turnover rate is experimentally determined without any delay by our MS-based approach because it directly provides access to the global rate of replacement of old by new material in the entire PG layer.

In our study, PG turnover was not found to be associated with the recycling of specific PG fragments, such as the tripeptide L-Ala-γ-D-Glu-DAP or glucosamine, as it occurs in *E. coli*[11,13,14] (Fig. 1). This observation does not rule out the possibility that PG moieties are recycled in metabolic pathways that are unrelated to PG synthesis since this would not result in hybrid muropeptide isotopologues[35,36]. PG recycling was previously explored based on the determination of MurNAc-6P in *B. subtilis* and *E. coli* mutants lacking the MurQ etherase for the conversion of MurNAc-6P into GlcNAc-6P, an obligate step for the recycling of MurNAc into the PG biosynthesis pathway (Supplementary Fig. 1B)[37]. In

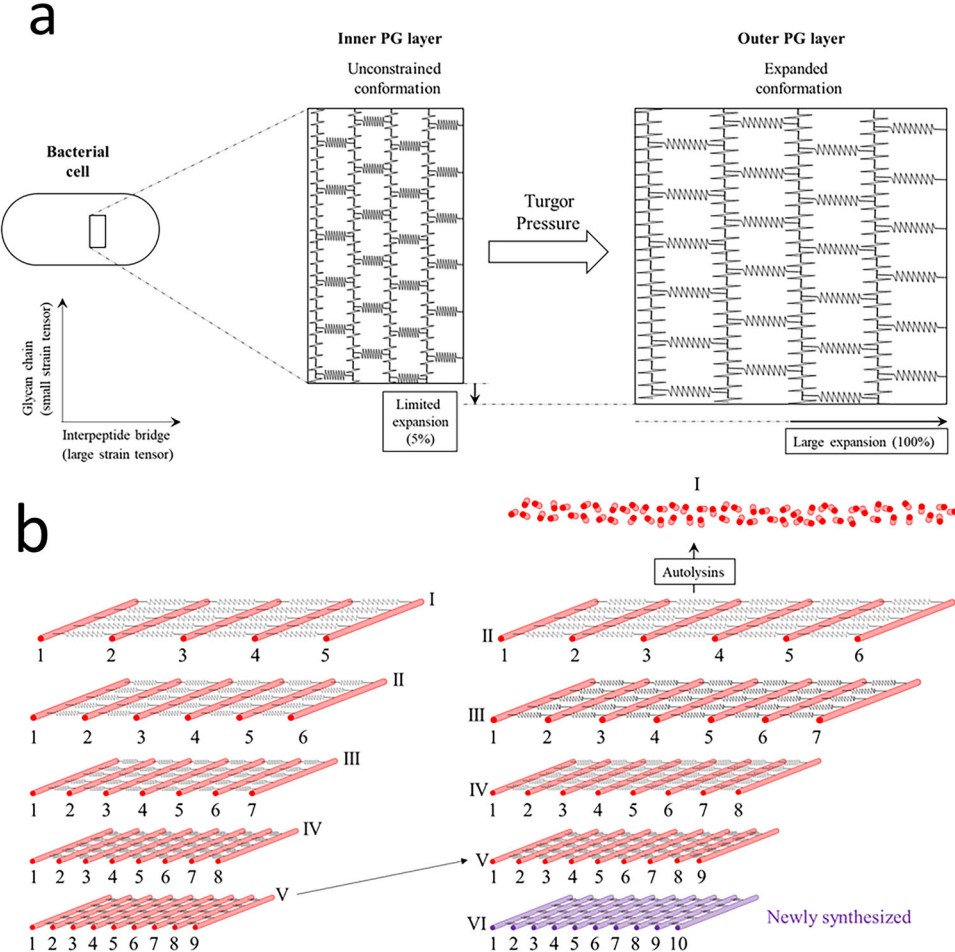

**Fig. 5 | Desquamation model of lateral cell wall expansion in *B. subtilis*. a** The turgor pressure applied to PG meshes mainly leads to lateral cell wall expansion because of the difference between the strain tensors of the peptide and glycan PG components. In the schematic, extensions of the PG mesh were arbitrarily set to 5% and 100% to illustrate the consequences of the difference in the strain tensors of the glycan chains and peptide bridges on the expansion of the PG mesh along, and perpendicularly to, the long axis of the cell, respectively. **b** Lateral expansion of the side wall upon incorporation of a newly synthesized mesh in the PG. PG layers are numbered using Roman numerals from the oldest to the newest. The right panel illustrates the insertion of a newly synthesized layer (VI, purple color) leading both to an upward shift in layers II-V (oblique arrow) and to the degradation of layer I. The Arabic numerals underscore the increasing number of glycan chains in the most recent PG layers. For the sake of clarity, only 5 out of the 10-30 PG layers are shown[1]. A difference of one glycan chain per layer is arbitrarily chosen along with a very large strain tensor to ease the expansion visualization.

the exponential phase of growth, the intracellular pool of MurNAc-6P was found to be 35-fold lower in *B. subtilis* than in *E. coli*, in spite of the presence of a 5-to-10-fold larger amount of PG per cell in the former species. These results indicate that only a minute proportion of the MurNAc originating from PG turnover is recycled in exponentially growing *B. subtilis*, in agreement with the absence of the corresponding hybrid isotopologues in the mass spectra acquired in our study.

## Methods

### Labeling of the *B. subtilis* PG with heavy ¹³C and ¹⁵N isotopes

PG labeling was performed in a complex medium consisting of the minimal M9 medium supplemented with 0.1% [$^{13}$C]glucose, 0.1% [$^{15}$N]NH$_4$Cl, and 0.1% labeled *E. coli* extract. The latter extract was prepared by growing *E. coli* BW25113 overnight in 6 liters of M9 minimal medium containing 0.1% [$^{13}$C]glucose and 0.1% [$^{15}$N]NH$_4$Cl. Bacteria were collected by centrifugation ($7500 \times g$ for 15 min at 4 °C), resuspended in 20 ml of water, and lysed by sonication. Bovine pancreas ribonuclease A (5 mg; Sigma) and DNase I (5 mg; PanReac AppliChem) were added and the lysate was incubated for 8 h at 37 °C. Proteinase K from *Tritirachium album* (5 mg; Euromedex) was added and the incubation was continued for 18 h at 37 °C. The extract was autoclaved at 120 °C for

15 min and centrifuged ($53,000 \times g$ for 30 min at 4 °C). The supernatant was lyophilized, weighted, and dissolved in 10 mL of sterile H$_2$O [final concentration of *ca.* 15% (weight/vol) or 0.15 g/mL]. The unlabeled *E. coli* extract was prepared by the same procedure except for the presence unlabeled glucose and NH$_4$Cl in the culture medium.

For PG labeling, *B. subtilis* strain 168 was grown overnight at the surface of a brain heart infusion (BHI) agar plate (Difco). One colony was inoculated in 250 mL of M9 minimal medium supplemented with 0.1% [$^{13}$C]glucose, 0.1% [$^{15}$N]NH$_4$Cl, and 0.1% labeled *E. coli* extract. Bacteria were grown at 37 °C with shaking (180 rpm) up to an optical density at 600 nm (OD$_{600}$) of 0.4. Bacteria from an aliquot of the culture (50 ml) were collected by centrifugation ($7500 \times g$ for 15 min at 4 °C) and the bacterial pellet was frozen at −65 °C. In parallel, the rest of the culture (200 ml) was centrifuged ($7500 \times g$ for 7 min at 25 °C). Bacteria were resuspended in prewarmed (37 °C) unlabeled M9 medium containing 0.1% glucose, 0.1% NH$_4$Cl, and 0.1% unlabeled *E. coli* extract. Incubation was continued at 37 °C with shaking (180 rpm). Aliquots collected 5, 10, 15, and 20 min after the medium switch were centrifuged ($7500 \times g$ for 15 min at 4 °C) and the bacterial pellets were frozen at −65 °C.

Switching the medium from labeled to unlabeled was chosen to facilitate the detection of neo-synthesized isotopologues by mass

spectrometry (see below). Indeed, these light isotopologues, which are present at a low abundance at the early times after the medium switch, have low $m/z$ values that do not overlap with those of the sodium adducts of uniformly labeled isotopologues, which are predominant at those early times.

### PG extraction and purification of muropeptides

The bacterial pellets from the 50-mL aliquots kept at −65 °C (above) were unfrozen by adding 10 mL of 4% SDS prewarmed at 100 °C. PG was collected by centrifugation (53,000 × $g$ for 10 min at 25 °C) and washed five times with water (10 mL). PG resuspended in 1 mL of 10 mM Tris-HCl (pH 7.5) was treated overnight at 37 °C with 200 µg/mL pronase (Sigma). PG was washed two times with 1 mL of water by centrifugation (21,000 × $g$ for 30 min at 25 °C). PG was treated overnight with 200 µg/mL trypsin (Sigma) in 1 mL of phosphate buffer (20 mM, pH 8.0), washed five times with 1 mL of water, and treated overnight with 200 µg/mL mutanolysin (Sigma) and 200 µg/mL lysozyme (Sigma) at 37 °C in 50 µL of 50 mM Tris-HCl (pH 8.0). Soluble disaccharide-peptides were reduced by the addition of 50 µL of 250 mM borate buffer (pH 9.0) containing 10 mg/mL sodium borohydride. The solution was incubated for 30 min at 25 °C, acidified (pH 4.0) with 20% (vol/vol) orthophosphoric acid, and centrifugated (12,000 × $g$ for 5 min at 25 °C). The supernatant containing the soluble disaccharide-peptide PG fragments (muropeptides) was extemporaneously subjected to $rp$HPLC.

For the separation of muropeptides by $rp$HPLC, the supernatant (100 µL) was injected into a $C_{18}$ column (Hypersil GOLD aQ 250 × 4.6, 3 µm; ThermoFisher) at a flow rate of 1.0 mL/min. A linear gradient (0% to 100%) was applied between 11.1 min and 105.2 min at 25 °C (buffer A: 0.1% TFA; buffer B: 0.1% TFA, 20% acetonitrile; vol/vol). The absorbance was monitored at 205 nm and muropeptides were collected in 1 mL-fractions. The muropeptides were lyophilized, solubilized in 20 µL of water, and stored at −20 °C.

### Analysis of muropeptides by mass spectrometry

The solubilized $rp$HPLC fractions containing the muropeptides (5 µL out of 20 µL) were injected into a Maxis II ETD mass spectrometer (Bruker) at a flow rate of 0.1 mL/min (50% acetonitrile, 50% water, 0.1% formic acid; vol/vol). Mass spectra were acquired in the positive mode with a capillary voltage of 3500 V, a pre-storage pulse of 18 µs, ion funnel 1 RF 300 Vp-p. Isotopologues and their relative abundance were identified as previously described[8]. The $m/z$ scan range was from 300 to 1850 at a speed of 2 Hz. Transfer time stepping was enabled with the following parameters: RF values 400 and 1200 Vp-p, transfer times 30 µs and 90 µs, timing 50 and 50%. Tandem mass spectra were obtained using a collision energy of 50 eV in the $m/z$ range of 150 to 1000 for muropeptide monomers and 150–2000 for muropeptide dimers with isolation width of 1. The applied collision energy was varied between 77 and 100% of the 50 eV setting with timings of 33% and 67%, respectively. Mass spectrometric data were analyzed using the free software mineXpert2[38].

### Preparation of the PG labeling reagent

Synthesis and characterization of ethinyl-D-Ala-D-Ala is reported in Supplementary Data 3.

### PG labeling and analysis of *B. subtilis* PG expansion by fluorescence microscopy

Three methods have been devised to decipher the mode of PG expansion by cell imagining. A first method is based on fluorescent vancomycin that specifically binds to the peptidyl-D-Ala-D-Ala extremity of pentapeptide stems. Because pentapeptide stems are the essential acyl donor substrate for the cross-linking reaction catalyzed by the D,D-transpeptidases and because acyl donors not used by the D,D-transpeptidases are rapidly converted to tetrapeptide stems in

most bacteria, fluorescent vancomycin labels bacterial cell wall regions containing the essential substrate for the cross-linking reaction catalyzed by these enzymes. Thus, fluorescent vancomycin only indirectly maps the sites of PG synthesis. A second method involves the use of fluorescent derivatives of D-Ala that can be incorporated either at the 5th position of pentapeptide stems or at the 4th position of tetrapeptide stems by an exchange reaction catalyzed by D,D-transpeptidases or L,D-transpeptidases, respectively. This method reveals the colocalization of catalytically active transpeptidases and their substrates, thus only providing an indirect mapping of the sites of PG synthesis. In this study, we used a third method aimed at providing a direct detection of the sites where new PG subunits are incorporated in the expanding side wall. To this end, D-Ala-D-Ala derivative containing an ethinyl group at the N-terminal end of the dipeptide are incorporated into cytoplasmic PG precursors in lieu of a small fraction of the endogenous D-Ala-D-Ala. This results in the presence of the ethinyl group at the 4th position of stem peptides in the periplasm. These groups specifically reacted with an azido derivative of a fluorescent molecule. This method enabled the direct detection of cell wall regions actively involved in PG synthesis.

In this study, cells of *B. subtilis* 168 were grown in BHI medium at 37 °C and 150 rpm until an $OD_{600nm}$ 0.4. A pre-warmed solution of ethinyl-D-Ala-D-Ala was then added to the cell culture to obtain a final concentration of 2 mM. Cells were further incubated for 15 min (37 °C, 150 rpm), pelleted (9000 × $g$, 5 min, 20 °C), resuspended in 1 × Dulbecco's-phosphate buffered saline (DPBS; Gibco) and fixed overnight in ice-cold 70% ethanol. Fixed cells were pelleted (9000 × $g$, 5 min, 20 °C), resuspended into DPBS containing 0.15% (vol/vol) Tween-20, and incubated for 5 min with gentle shaking. The cells were then washed once in DPBS containing 1% (weight/vol) bovine serum albumin (BSA) before coupling to azido-AF532 (32 µM; Click Chemistry Tools) during 45 min, using the Click-iT™ cell reaction buffer kit (Invitrogen). The sample was washed twice in DPBS before mounting in DPBS for fluorescence microscopy.

For phase contrast, bright-field, and conventional fluorescence microscopy, samples were mounted between a microscope slide and a coverslip, and observed at 20 °C using a motorized two-deck Olympus IX83 optical microscope equipped with a UPFLN 100X O-2PH/1.3 objective and an ORCA-Flash4.0 Digital sCMOS camera from Hamamatsu. Images were acquired using the Volocity software package. For quantitative comparisons of fluorescence intensity, images were acquired with identical illumination and acquisition parameters.

### Reporting summary

Further information on research design is available in the Nature Portfolio Reporting Summary linked to this article.

## Data availability

Source data are provided with this paper. The microscopy datasets generated in this study have been deposited in the Zenodo repository under accession number 15488100.

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

## Acknowledgements

We thank R. Puppo and A. Marie for technical assistance in the collection of mass spectra data at the Plateau Technique de Spectrométrie de Masse Bio-Organique of the Muséum National d'Histoire Naturelle. We thank C. Anoyatis-Pelé for critical reading of the manuscript. Support for this work comes from the Agence Nationale de la Recherche (ANR-23-CE11-0029 to CM and YSW; ANR-24-CE11-7056 to MA, CM, and YSW; Labex ARCANE ANR-17-EURE-0003 to YSW). This work used the plat-forms of the Grenoble Instruct-ERIC Center (ISBG; UMS 3518 CNRS-CEA-UGA-EMBL) within the Grenoble Partnership for Structural Biology (PSB), supported by FRISBI (ANR-10-INBS-05-02) and GRAL, financed within the University Grenoble Alpes graduate school (Ecoles Universitaires de Recherche) CBH-EUR-GS (ANR-17-EURE-0003). IBS acknowledges inte-gration into the Interdisciplinary Research Institute of Grenoble (IRIG, CEA).

## Author contributions

Y.L. conceived and designed the project, acquired, analyzed, and interpreted data, and drafted the article. L.B. and C.M. performed the microscopy and labeling experiments and analyses, interpreted the data,a and drafted the respective sections of the article. Y-S. W. syn-thesized the ethinyl-D-Ala-D-Ala molecule. J-E.H., F.R. and M.A. con-ceived and designed the project, analyzed and interpreted data, and drafted the article.

## Competing interests

The authors declare no competing interests.
