## [Peer Review file · Nature Communications]

Mechanism of lateral cell-wall expansion at a constant diameter in *Bacillus subtilis*

Corresponding Author: Dr Michel Arthur

Version 0:

Reviewer comments:

Reviewer #1

(Remarks to the Author)
Review

In the ms by Liang et al., the authors investigate the mechanism of lateral cell wall expansion in the Gram-positive model bacterium *Bacillus subtilis*. Expansion of the cell wall during growth in the Gram-negative model bacterium *E. coli* occurs by insertion of newly formed peptidoglycan strands into the existing cell wall network. This view of cell wall expansion, however, is questioned for Gram-positive bacteria, e.g. since cell-wall-less L-forms are able to recover a peptidoglycan cell wall, while lacking a template matrix. The authors apply a recently developed, nifty heavy-isotope labeling strategy in combination with mass spectrometry as well as a novel cell wall labeling and imaging approach to provide evidence for a mechanism of lateral cell wall expansion via assembly of a new peptidoglycan mesh that is not connected to the older meshes, which instead is outward shifted and eventually degraded. Thereby showing that *E. coli* (Gram-) and *B. subtilis* (Gram+) have very distinct modes of peptidoglycan assembly. Their data support the generally accepted outward growth model of *B. subtilis*.

This is a very solid piece of work. The ms is clearly written, technically sound and provides an advancement to the scientific field. The approaches are sophisticated and well-conducted and their conclusion are mostly justified.

The authors first used the labeling strategy to determine an estimated 69% PG degradation in one generation and provided evidence for a recycling pathway distinct from *E. coli*. They further show labeled Glu incorporation in the peptidoglycan that may originate from existing pools. Isotopic composition of dimers indicate that new peptidoglycan chains are exclusively crosslinked with each other, indicating the pre-formation of new material and the co-occurring turnover of old material in the outer cell wall zone, in *B. subtilis*, whereas new peptidoglycan stains are predominantly connected to old peptidoglycan in *E. coli*, indicating a coordinated insertion and degradation. Zonal expansion in the peripheral area of the septum, as described in *Streptococcus pneumoniae*, is refuted in *B. subtilis* on the basis of disperse incorporation of new peptidoglycan material.

Major points:

The expansion model presented in Figure 5 seems to neglect the very likely occurring gradual cleavage of the expanding network by autolysins. Instead it suggests a rather unlikely instantaneous autolytic cleavage of the maximally expanded cell wall.

Most critical is the interpretation of the hybrid/mixed isotopologues. The MS data (Fig. 1) suggest that a great variation of isotopologues with different mass shifts occur, and not only the mentioned +5 +6 shifts, which apparently are slidely over-representated. So I see not justification in refuting the assumption that incorporation of various amino acids/amino sugars occur through the recycling ?

Minor points:

L35: it is not clear to me on which basis the molecular mass of $10e10 - 10e11$ Da of the peptidoglycan was calculated ?

Figure 1: the two colors of red are hard to distinguish

Extended data Figure 1.: symbols in hybrids figure 1B are much larger than in 1A and partially overlapping.

Reviewer #2

(Remarks to the Author)

Cell wall synthesis is a vital process for bacteria, as the cell wall provides shape and structure to the cell. While the thickness of the cell wall is different in Gram-negative and Gram-positive organisms, the subunits that make up the wall are conserved. Does this mean that the wall is built in the same manner? The authors have previously used a labelling and mass spec protocol to determine how new subunits are inserted into the Gram-negative *E. coli* cell wall. Here they use a similar method to determine how new subunits are inserted in the Gram-positive *B. subtilis*. Interestingly, they find that the mode of insertion is different for these two rod shaped organisms including the formation of tripeptide and tetra peptides as well as the timing of transglycosylation and transpeptidation reactions.

Major

Overall, the manuscript would be difficult for someone who is not an expert in cell wall synthesis/recycling to understand. The introduction should include more information about PG recycling and the byproducts of this, perhaps with a figure showing the structure and where cleavage events take place and the resulting structures. The authors nicely explain how tetra and tri peptides are produced in line 136-139 yet this knowledge would have been useful to understand figs 1-2. I do not think 2A is clear enough to distinguish the fragments and where the cleavage events take place.

- 1) Table 1: How does this show that labeled and unlabeled are in the same amount. I don't see anything indicating labeling
- 2) The t50% for bacillus is commented to be shorter than a generation time, yet the *e coli* data shows *e coli*'s is longer than a generation time. The authors should comment on what this means. Is it a result of the tetra and tri peptide t50 being different?
- 3) PMC7308169 AFM has been done bacillus cells and revealed different PG architecture depending on how deep you go. How does this relate to the desquamation model?

Minor

- 1) Missing "introduction" as section title
- 2) growth in an *e coli* extract is unusual and should be commented on in the text without making the reader find online supp material.
- 3) Table 1 is mentioned before defining what t50% is. Making it difficult to understand the table.
- 4) The legend for figure 1 says fully unlabeled, is this supposed to mean T20? True unlabeled should be a strain that was never grown with the isotopes.
- 6) why are the masses of bacillus and *e coli* tetra different in fig 1?
- 5) line 131 is missing a period
- 6) Fig 3A: please explain what the * is A. I find B to be quite confusing. What is the dimer-monomer? Is there an x axis or are you just spreading the data so we can see. Maybe it would be clearer if you had a bar graph or some other graph and grouped the dimer with its partner monomer
- 7) Fig 4: A) is this necessary, B) when printed I only see septal insertion and this not mentioned in the text.

Reviewer #1:

Major points:

The expansion model presented in Figure 5 seems to neglect the very likely occurring gradual cleavage of the expanding network by autolysins. Instead it suggests a rather unlikely instantaneous autolytic cleavage of the maximally expanded cell wall.

Answer: Figure 5 schematically represents the proposed mechanism underlying PG turnover. It is not meant to imply that the outermost layer is uniformly and instantaneously turned over across the entire cell surface or that underlying layers do not undergo any cleavage at all. In fact, such cleavage events have been inferred from atomic force microscopy studies (PMC7308169), which revealed the presence of PG pores (see also answer #3 to reviewer #2). However, because our experimental approach does not enable the detection of pore formation, we believe it is appropriate to retain the current schematic, which emphasizes the conceptual framework of our model rather than fine-grained spatial or temporal resolution of cleavage events.

Most critical is the interpretation of the hybrid/mixed isotopologues. The MS data (Fig. 1) suggest that a great variation of isotopologs with different mass shifts occur, and not only the mentioned +5 +6 shifts, which apparently are slidely over-representated. So I see not ?

Answer: The additional peaks pointed out by the reviewer include the sodium and potassium adducts of the unlabeled species ($m/z = 482.214$ and 490.197 , respectively). Fig. 1 was modified to label these peaks in the mass spectra. Additional low-intensity peaks do not correspond to any of the recycling products found in *Escherichia coli*, including a peak at $m/z = 482.250$, which could not be assigned to any isotopologue of the amidated disaccharide-tetrapeptide.

a
Hybrid isotopologues

**b**
Hybrid isotopologues

*E. coli* BW25113

Minor points:

L35: it is not clear to me on which basis the molecular mass of 10^{10} – 10^{11} Da of the peptidoglycan was calculated?

Answer: Wientjes et al. (J Bacteriol 1991 173) determined the PG content of exponentially growing *E. coli* using radiolabeled DAP. This approach provided a number of disaccharide-peptide PG units per cell of $3.5 \cdot 10^6$. Taking an average mass of 900 Da per disaccharide unit, the molecular weight of an *E. coli* PG sacculus is thus $3.2 \cdot 10^9$ Da. For *Bacillus subtilis*, an estimate of $3 \cdot 10^{10}$ Da can be proposed considering that the *B. subtilis* PG is about 10 times thicker than that of *E. coli* (1-3 versus 10-30 layers), hence the proposed range of 10^{10} to 10^{11} Da in the introduction of the manuscript.

Figure 1: the two colors of red are hard to distinguish.

Answer: This color code has been used in previous publications (purple and red for new and old PG structures, respectively). For consistency and to aid reader comprehension, we believe it is helpful to maintain this color code in the present report.

Extended data Figure 1: symbols in hybrids figure 1B are much larger than in 1A and partially overlapping.

Answer: We do agree with the reviewer that the symbols are larger in 1B than in 1A. This was corrected in the revised version of the figure.

Reviewer #2:

Major

Overall, the manuscript would be difficult for someone who is not an expert in cell wall synthesis/recycling to understand. The introduction should include more information about PG recycling and the biproducts of this, perhaps with a figure showing the structure and where cleavage events take place and the resulting structures. The authors nicely explain how tetra and tri peptides are produced in line 136-139 yet this knowledge would have been useful to understand figs 1-2. I do not think 2A is clear enough to distinguish the fragments and where the cleavage events take place.

Answer: We do agree with the reviewer and have expanded the first paragraph of the introduction to include more information on PG synthesis, maturation, and recycling. The pathways are described in a new figure (Supplementary Fig. 1) as shown below.

A

B

Supplementary scheme 1. Peptidoglycan assembly and maturation reactions.

A. Periplasmic biosynthetic and maturation steps in *B. subtilis*. Enzymes involved in biosynthetic and maturation reactions are shown in blue and red, respectively. The PG is assembled from a disaccharide-pentapeptide subunit (highlighted in grey). The green double arrow indicates a cross-link between stem peptides. The black double arrow indicates the bond cleaved by amidases. Glycosyltransferases polymerize glycan strands by forming β -1 \rightarrow 4 glycosidic bonds. Lytic glycosyltransferases cleave the MurNAc-GlcNAc bond and form an internal anhydro bond, producing MurNAc^{Anh}. DAP_{NH2}, amidated diaminopimelic acid.

B. Overall biosynthesis of *E. coli* PG and recycling pathways. Cross-linked PG is cleaved by lytic transglycosylases, amidases, and endopeptidases, generating peptides and GlcNAc-MurNAc^{Anh}-peptide fragments that are transported into the cytoplasm by the Opp and AmpG permeases, respectively. Two moieties are recycled: (i) the tripeptide L-Ala-D-iGlu-DAP, which is added to UDP-MurNAc by the dedicated recycling enzyme Mpl and (ii) the glucosamine (GlcN) moiety of GlcNAc and MurNAc, which is recycled in the neosynthesis of both of these sugars. Of note, the acetyl group (Ac) of GlcNAc and MurNAc and the D-lactoyl groups of MurNAc do not originate from recycling, the latter being indicated by partial red coloring of the residue name. The complete disaccharide-pentapeptide subunit, linked to the

undecaprenyl-lipid carrier via a pyrophosphate bond, is highlighted in grey. Synthesis of the disaccharide-pentapeptide subunit involves the production of UDP-MurNAc from UDP-GlcNAc by MurA and MurB, followed by sequential addition of L-Ala, D-Glu, and *meso*DAP to form UDP-MurNAc-L-Ala-D-iGlu-DAP. MurF then adds the D-Ala-D-Ala dipeptide to form UDP-MurNAc-pentapeptide. This precursor is transferred to the undecaprenyl lipid carrier (blue rectangle in the cytoplasmic membrane), followed by the addition of GlcNAc.

1) Table 1: How does this show that labeled and unlabeled are in the same amount. I don't see anything indicating labeling.

Answer: The $t_{50\%}$ values reported in the "Average" column of the table were calculated from four independent kinetics experiments. Typical graphs of these experiments are shown in Fig. 2. Individual $t_{50\%}$ values for the four experiments were determined as the time at which the two curves (red and purple) intersect. Therefore, at this time point, the relative amounts of labeled and unlabeled muropeptide isotopologues are equal. We have added explanatory text in the legend to Fig. 2 and in Table 1.

2) The $t_{50\%}$ for bacillus is commented to be shorter than a generation time, yet the e coli data shows e coli's is longer than a generation time. The authors should comment on what this means. Is it a result of the tetra and tri peptide t_{50} being different?

Answer: The generation times in minimal medium were 67 ± 1 min and 36 ± 5 min for *E. coli* (Atze *et al.* 2022, reference 9) and *B. subtilis* (this report), respectively. Thus, the $t_{50\%}$ values are smaller than the generation times for both *E. coli* and *B. subtilis*. These values have been specified in a sentence at the end of the legend to Fig. 2.

3) PMC7308169 AFM has been done bacillus cells and revealed different PG architecture depending on how deep you go. How does this relate to the desquamation model?

Answer: In this remarkable publication, atomic force microscopy (AFM) experiments revealed that the mature surface of live cells exhibits a landscape punctuated by large (up to 60 nm in diameter) and deep (up to 23 nm) pores. While these findings provide valuable structural insights, the described architecture does not inform on the rate and mechanism of PG turnover. Conversely, our current study is not designed to assess the contribution of pore formation to PG dynamics. Nonetheless, we do not believe that the existence and morphology of these pores challenge the desquamation model, which proposes an outward shift of PG layers, a model that remains compatible with the AFM-detected pores reported in PMC7308169. Given the distinct objectives and methodology of our study, we believe that a discussion of the pore contribution to PG dynamics falls outside the scope of the present manuscript and we would prefer not to include it.

Minor

1) Missing "introduction" as section title.

Answer: We have added the section title as requested.

2) growth in an e coli extract is unusual and should be commented on in the text without making the reader find online supp material.

Answer: We have added two sentences in the relevant paragraph to briefly describe the extract preparation (Lines 80-82).

3) Table 1 is mentioned before defining what t50% is. Making it difficult to understand the table.

Answer: This point was addressed in the answer to one of the major points (point 1, related to Table 1, above). We hope that the modifications to the footnote to Table 1 and the legend to Fig. 2 introduced in the revised text in response to that major comment also correctly address the present comment. We have also mentioned Fig. 2a before Table 1 (Lines 93-94).

4) The legend for figure 1 says fully unlabeled, is this supposed to mean T20? True unlabeled should be a strain that was never grown with the isotopes.

Answer: True unlabeled material is included in Fig. 1. The mass spectrum at the top of panel A of Fig. 1 (t = 0 min) only contains mass peaks corresponding to labeled material. For this time point, bacteria were collected just before the heavy-to-light medium switch. T20 refers to the sample collected 20 min after the medium switch. The title of the figure now reads: “Mass spectra showing the labeled, unlabeled, and hybrid isotopologues” to better indicate the labeling status of the observed isotopologues. Furthermore, we have added an explanatory sentence to formally define the notion of “labeled”, “unlabeled”, and “hybrid” isotopologues. We took advantage of this comment to remove “fully” from the figure title to be consistent with the whole text.

5) line 131 is missing a period

Answer: Fixed.

6) why are the masses of bacillus and e coli tetra different in fig 1?

Answer: The masses differ because DAP residues are amidated in *B. subtilis* but not in *E. coli*. This is indicated at lines 39 to 41 of the revised manuscript.

7) Fig 3A: please explain what the * is A. I find B to be quite confusing. What is the dimer-monomer? Is there an x axis or are you just spreading the data so we can see. Maybe it would be clearer if you had a bar graph or some other graph and grouped the dimer with its partner monomer

Answer: The star (*) is actually a multiplication symbol to indicate the number of amidated DAP residues in the dimers.

The reviewer is correct. The abscissae axis of the graph appearing in panel B of Fig. 3 has no label. This presentation is used to spread the data so as to avoid overlaps of the corresponding symbols.

The presentation of the data in panel B has been chosen to ease the statistical evaluation of the difference between the $t_{50\%}$ of monomers and dimers containing the same free peptide stem. We have added an explanatory sentence and an example in the legend to specify how the $\Delta t_{50\%}$ (dimer – monomer) was calculated.

8) Fig 4: A) is this necessary, B) when printed I only see septal insertion and this not mentioned in the text.

Answer: We would like to maintain Fig. 4 because it is critical to our data analysis strategy, which combines the localization pattern of PG synthesis during cell elongation (uniform labeling of the side wall) with the mode of PG expansion (newly synthesized stems cross-linked to newly synthesized stems). The fluorescence signal along the side walls is clearly visible when the figure is viewed on screen. We will ensure that this pattern is preserved and clearly visible in the final proofs.